# Study of the Antibacterial Activity of Superhydrophilic and Superhydrophobic Copper Substrates against Multi-Drug-Resistant Hospital-Acquired *Pseudomonas aeruginosa* Isolates

**DOI:** 10.3390/ijms25020779

**Published:** 2024-01-08

**Authors:** Natalia E. Bondareva, Anna B. Sheremet, Elena Y. Morgunova, Irina R. Khisaeva, Alisa S. Parfenova, Marina Y. Chernukha, Fadi S. Omran, Alexandre M. Emelyanenko, Ludmila B. Boinovich

**Affiliations:** 1Department of Medical Microbiology, Gamaleya National Research Center for Epidemiology and Microbiology, Ministry of Health of the Russian Federation, 18 Gamaleya St., 123098 Moscow, Russia; nataliia.d@mail.ru (N.E.B.); anna-pimenova@mail.ru (A.B.S.); lena.morgunova.1968@mail.ru (E.Y.M.); irakhisaeva2708@gmail.com (I.R.K.); alisaparfenova6@gmail.com (A.S.P.); chernukha08@mail.ru (M.Y.C.); 2A.N. Frumkin Institute of Physical Chemistry and Electrochemistry, Russian Academy of Sciences, Leninsky Prospect 31, 119071 Moscow, Russia; duckyfriedrich@gmail.com (F.S.O.); ame@phyche.ac.ru (A.M.E.)

**Keywords:** multidrug resistance, *P. aeruginosa*, superhydrophilicity, superhydrophobicity, nosocomial infections

## Abstract

The global spread of multidrug-resistant (MDR) hospital-acquired pathogens is a serious problem for healthcare units. The challenge of the spreading of nosocomial infections, also known as hospital-acquired pathogens, including Pseudomonas aeruginosa, must be addressed not only by developing effective drugs, but also by improving preventive measures in hospitals, such as passive bactericidal coatings deposited onto the touch surfaces. In this paper, we studied the antibacterial activity of superhydrophilic and superhydrophobic copper surfaces against the *P. aeruginosa* strain PA103 and its four different polyresistant clinical isolates with MDR. To fabricate superhydrophilic and superhydrophobic coatings, we subjected the copper surfaces to laser processing with further chemosorption of fluorooxysilane to get a superhydrophobic substrate. The antibacterial activity of superhydrophilic and superhydrophobic copper surfaces was shown, with respect to both the collection strain PA103 and polyresistant clinical isolates of *P. aeruginosa*, and the evolution of the decontamination of a bacterial suspension is presented and discussed. The presented results indicate the promising potential of the exploitation of superhydrophilic coatings in the manufacture of contact surfaces for healthcare units, where the risk of infection spread and contamination by hospital-acquired pathogens is extremely high.

## 1. Introduction

Nosocomial infections, also known as hospital-acquired infections (HAIs), are a global problem in hospitals, particularly intensive care units (ICUs), surgical units, burns units, etc. Complications of nosocomial infections result in a poorer prognosis for the underlying disorder, longer hospital stays, worsening of patients’ quality of life, higher costs of treatment, and increased mortality [1,2,3,4]. In the structure of hospital lethality, HAI ranks fourth after diseases of the cardiovascular system, malignant tumors, and acute cerebrovascular accidents. The mortality rate for patients with HAI is almost two to three times higher than for similar groups without this complication [5].

*Pseudomonas aeruginosa* is a gram-negative anaerobic pathogen that is still one of the most important causative agents of nosocomial infections. *P. aeruginosa* accounts for about 10% to 11% of HAI cases. This high prevalence in hospitals is due to the acquired resistance of this pathogen to a large number of antibacterial drugs and disinfectants. *P. aeruginosa* causes respiratory, urinary, and wound infections in hospitals and can cause bacteremia, particularly in patients in ICUs and on mechanical ventilators [1]. Due to the acquisition of multidrug resistance (MDR), treatment of the *P. aeruginosa* HAI is difficult [6].

One of the areas of the fight against nosocomial infections is prophylaxis measures, to which, in addition to the standard precautions of staff, the hygiene of hospital surfaces can be attributed. For example, it is known that various surfaces of the hospital, including porous surfaces (beds, mattresses, etc.) and non-porous surfaces (bed rails, bells, switches, door handles), are prone to microbial contamination and can serve as a source of preservation and dissemination for hospital-acquired pathogens [7].

One way to combat the spread of nosocomial infections through prolonged exposure to contact surfaces may be to manufacture these surfaces from microbicidal materials. Copper is known to be one of these materials [8,9,10,11,12,13,14]. The antimicrobial activity of copper has been studied in a number of clinical studies; however, prolonged contact with copper surfaces can lead to negative effects on the human body. For example, excess copper entering the body can lead to pathological conditions [15].

In order to prevent the penetration and accumulation of toxic copper nanoparticles in the human body, it is possible to use coatings with extreme wettability, in which the copper nanoparticles are firmly attached to the substrate. The extreme wetting coatings correspond to either superhydrophilic or superhydrophobic surfaces with a hierarchical roughness composed of the aggregates of nanoparticles on the surface. Such coatings contribute not only to the preservation of the bactericidal properties of copper but also to their increase due to the highly developed surface area [15]. This is related to the enhanced release of copper ions in contact with the aqueous medium.

In this work, we studied the antibacterial activity of superhydrophilic (SHPhil) and superhydrophobic (SHPhob) copper surfaces against both collection strains and polyresistant clinical isolates of *P. aeruginosa.* Four protocols, as specified in Section 3, “Materials and Methods”, were used to assess the bactericidal activity of the studied surfaces at different temperatures.

## 2. Results and Discussion

### 2.1. Study of the Antibacterial Activity of Bare, Superhydrophobic, and Superhydrophilic Copper Substrates Placed in a Liquid Nutrient Medium at Different Temperature Regimes

At the first stage, the study of the antibacterial activity of bare, superhydrophobic, and superhydrophilic copper substrates in a liquid nutrient medium towards the *P. aeruginosa* PA103 isolate was carried out at room temperature after 1, 4, 24, 96, 120, 144, 168, 216, and 312 h. It was found that 96 h after the application of copper substrates and in the control sample, completion of bacterial logarithmic growth and transition to the stationary phase and subsequently to the extinction phase were shown. Furthermore, the application of superhydrophobic and superhydrophilic copper substrates caused a shift in the completion of logarithmic growth and a transition to the stationary phase 24 h earlier than in the reference sample. It is also worth noting that, in addition to reducing the logarithmic growth time of the culture, the introduction of superhydrophobic and superhydrophilic copper substrates led to an acceleration of the extinction phase. Thus, after 312 h, the bacterial titer was decreased to ~2 × 10^6^ CFU/mL for superhydrophobic and 1.7 × 10^5^ CFU/mL for superhydrophilic copper substrates introduced into the culture and to a titer of 3.25 × 10^8^ for the positive control. Reference bare copper substrates also showed a decrease in bacterial titer to 6 × 10^7^ CFU/mL (Figure 1).

Subsequently, experiments were carried out to study the bactericidal activity of bare, superhydrophobic, and superhydrophilic copper substrates against the same *P. aeruginosa* PA103 strain at 24, 96, 144, 216, and 312 h, but at 30 °C. The temperature of 30 °C was chosen because the range from 30 to 37 °C is optimal for *P. aeruginosa* growth.

It has been shown that when a bacterial suspension is cultivated in the presence of the plate samples under investigation at a temperature closer to the optimum temperature for the cultivation of pseudomonas, the antibacterial activity of bare, superhydrophobic, and superhydrophilic copper substrates is less pronounced (Figure 2).

As shown in Figure 2, the increase in the number of bacteria at 30 °C lasted up to 96 h in all samples tested. At 216 h, the growth of pseudomonas with the addition of reference copper substrates did not differ from the growth in the control dispersion. The addition of superhydrophobic or superhydrophilic copper substrates reduced bacterial titers to 4 × 10^7^ CFU/mL, compared with 7.2 × 10^8^ CFU/mL for the positive control. However, the greatest antibacterial activity for both superhydrophilic and superhydrophobic copper coatings was observed 144 h after the beginning of the experiments.

### 2.2. Study of the Antibacterial Activity of Bare, Superhydrophobic, and Superhydrophilic Copper Substrates When Depositing the Culture to the Surface of the Plates

Due to the fact that superhydrophilic and superhydrophobic coatings were applied to the test samples on one side, and the immersion of the plates in bacterial suspension could not exclude the contribution of untreated copper to the bactericidal effect, the modified protocol described by Boinovich et al. [15] was reproduced.

At the first stage, the antibacterial activity of bare, superhydrophobic, and superhydrophilic copper substrates was studied against the same *P. aeruginosa* PA103 strain as used in the first protocol experiments.

As can be seen in Figure 3, in the positive control sample, the bacterial titer remained virtually unchanged and ranged from 1.1 × 10^6^ CFU/mL when applied to the plates to 2.7 × 10^6^ CFU/mL 6 h after deposition. When applying a 10 μL bacterial suspension to a superhydrophobic plate, it was noted that, due to the properties of the plate and the humid conditions created in the covered Petri dish, the bacterial droplets were left on the surface without drying out. For superhydrophobic copper surfaces, the bacterial titer after 6 h was 4.8 × 10^6^ CFU/mL, which was comparable to the positive control. Deposition of the bacterial suspension to the reference copper plates showed their marked antibacterial activity, as, 2 h after deposition, the bacteria could not be incubated. For superhydrophilic copper coatings, complete growth suppression was observed already after 30 min.

In the above-described experiments, antibacterial activity was assessed using a laboratory strain of *P. aeruginosa* PA103. Our next task was to study the antibacterial activity of the plates under investigation towards the *P. aeruginosa* isolates that circulate in hospitals, and which are extremely problematic due to their multiple antibiotic resistance, causing hospital-acquired infections. To target this aim, we used the second protocol.

An evaluation of the antibacterial activity of the studied plates against the clinical isolates of *P. aeruginosa* showed that, in the presence of bare copper and superhydrophilic copper coatings, effective suppression of bacterial growth was observed. However, the suppression dynamics were slightly different for different isolates. The most pronounced effect was observed for copper plates against PA52Ts17 isolates, where the complete suppression of growth occurred after 15 min. The longest time for growth suppression was 2 h for the PAKB6/2014 isolate. For the superhydrophobic copper coating, no growth reduction was observed within 3 h of contact for any isolates, and the growth curves were not different from those for the control culture. Therefore, bare copper and superhydrophilic copper coatings have high antibacterial activity against both the laboratory strain and various *P. aeruginosa* clinical isolates with antibiotic resistance, with a very fast-acting antibacterial effect from 15 min to 2 h, resulting in complete suppression of pseudomonas growth in contact with the plates (Figure 4).

### 2.3. Study of the Antibacterial Effects of Bare, Superhydrophobic, and Superhydrophilic Copper Substrates towards P. aeruginosa PA103

At the next stage, we were faced with the task of determining the nature of the antibacterial action of the bare, superhydrophobic, and superhydrophilic copper substrates under study against *P. aeruginosa* PA103. A third protocol was used for this purpose.

In accordance with the protocol, *P. aeruginosa* PA103 was cultivated in PBS for 24 h at room temperature, after which the culture was incubated on Petri dishes containing cetrimide agar. In the experiment, the effects of different initial amounts of bacterial cells, 10^5^ and 10^6^ CFU/mL, were evaluated (Table 1).

It was shown that when *P. aeruginosa* PA103 was cultivated with the coatings under study in PBS, which did not contain nutrients for the optimal growth of bacteria, there was a pronounced antibacterial effect of the plates in comparison with the control. For example, when cultivating plates in 1 mL of PBS containing 10^5^ CFU/mL of *P. aeruginosa* PA103, antibacterial activity was shown for all surfaces under investigation. However, as the titer of pseudomonas increased and the plates were cultured in a suspension containing 10^6^ CFU/mL of *P. aeruginosa* PA103, the antibacterial effects of the superhydrophobic copper coatings decreased (Table 1).

In order to determine the nature of the antibacterial action of the copper substrates under investigation and to detect the possible formation of uncultivated forms of P. aeruginosa, an aliquot culture of 100 μL from each specimen was transferred after 24 h of contact to the fresh culture medium and incubated under standard conditions at 37 °C.

Bare copper and superhydrophilic coatings were found to be the most effective in inhibiting bacterial viability in experiments with a lower dose of pseudomonas. When exposed to high doses, complete viability inhibition could not be achieved under the experimental conditions because single cells could subsequently reproduce (Table 2). These results show the antibacterial activity of the studied coatings, for each of which the most effective modes of contact with pathogenic bacteria should be characterized in the future. Based on these findings, it can be assumed that contact with the substrates does not result in the formation of long-existing uncultivated forms, which could then spread and become vegetative forms.

### 2.4. Identification of Viable Forms of P. aeruginosa PA103 after Cultivation in the Presence of Bare, Superhydrophobic, and Superhydrophilic Copper Substrates

The antibacterial action shown had to be characterized in relation to the bactericidal or bacteriostatic mechanism of growth suppression. For this purpose, at this stage of the research, the method of morphological assessment of the structure of the bacteria and the fluorescent method for assessment of the integrity of the cell wall were applied. The commercial Live/Dead Viability/Cytotoxicity Kit was used, which made it possible to differentiate the viable cells with specific green light as a result of intracellular esterase detection from dead cells, into which, due to cell wall disturbance, ethidium bromide penetrates intracellularly and stains the DNA with red light. For these experiments, a fourth protocol was used, suggesting incubation of the investigated plate samples in a nutrient medium, after the preliminary application of 0.01 mL of bacterial *P. aeruginosa* according to the second protocol.

As shown in Figure 5, the positive control sample was dominated by viable forms of *P. aeruginosa* PA103 and only a few bacteria were non-viable, which is typical of a normal bacterial population. Exposure to all kinds of copper substrates resulted in virtually complete suppression of the viability of pseudomonas, which was due to disruption of the cell wall. The loose morphology of the cells deposited onto the superhydrophilic surface, analyzed by scanning electron microscopy, is shown in Figure 5e. Microscopic evaluation showed that the bacterial cells undergo lysis after contact with the coatings, as far as atypical pseudomonads with loose cell wall structure were observed in the preparations. The results obtained make it possible to draw a conclusion about the bactericidal action of the surfaces under investigation.

## 3. Materials and Methods

### 3.1. Surfaces Studied

The experimental copper samples, with sizes of 10 × 10 × 1 mm^3^, were cut from soft copper rolled sheets of grade M1M. According to the manufacturer’s data, the elemental composition of the M1M alloy is (in weight %): Cu 99.9, Fe 0.005, Ni 0.002, S 0.004, As 0.002, Pb 0.005, Zn 0.004, Ag 0.003, O 0.05, Sb 0.002, Bi 0.001, Sn 0.002.

The surface treatment was performed with the laser system “Argent-M” (Russia) with an IR-ytterbium fiber laser (wavelength 1064 μm), equipped with a unit for 2-axis deviation of the laser beam. Before the laser treatment, samples were processed in an ultrasonic bath with deionized water and dried in air. The laser treatment was carried out in an oxygen-enriched atmosphere at a temperature of 20–25 °C.

Single-pass raster scanning with a linear speed of 100 mm/s with parallel line increments of 0.0025 mm, a pulse length of 200 ns, a repeat frequency of 20 kHz, and a peak power of 0.95 mJ in TEM_00_ mode ensured that the samples achieved the roughness necessary to obtain a superhydrophilic and superhydrophobic state (SHPhil and SHPhob, respectively). The laser beam focused on the surface of the specimen in a spot of 40 μm in diameter (level 1/e^2^) with an energy density of ≈19 J/cm^2^. Immediately after laser treatment, the resulting surfaces were superhydrophilic (apparent contact angle ~0°), as evidenced by the rapid complete spreading of water droplets on the surface and their absorption. In order to produce superhydrophobic surfaces, the laser-treated samples were additionally exposed to the saturated vapors of methoxy-{3-[(2,2,3,3,4,4,5,5,6,6,7,7,8,8,8 pentadecafluoroctyl)-oxy]-propyl}-silane for 1 h at T = 95 °C. Subsequent drying for 1 h in a furnace at 130 °C resulted in the formation of a cross-linked layer of hydrophobic agent with low surface energy and the achievement of a superhydrophobic state at the laser-treated surface. The contact angle for the as-prepared superhydrophobic surface was 170.7 ± 0.25°, and the roll-off angle was 2.6 ± 1.2°. Non-laser-treated M1M plates (bare copper) were used as reference samples and had a contact angle of 71.1 ± 8.9° [16].

The antibacterial activity of superhydrophobic, superhydrophilic, and reference copper substrates was studied with respect to the *P. aeruginosa* strains and isolates listed in Table 3.

### 3.2. Study of the Antibacterial Activity of Bare, Superhydrophobic, and Superhydrophilic Copper Substrates

The study of the antibacterial activity of bare copper, superhydrophilic, and superhydrophobic coatings was carried out at room temperature and 30 and 37 °C. Before the test, the copper plates were sterilized under UV light in a Petri dish for 30 min on each side.

The bacterial culture of *P. aeruginosa* was grown on a Luria-Bertani (LB) broth (BD Difco™) liquid medium at 37 °C for 18 h. The *P. aeruginosa* colony-forming unit/mL (CFU/mL) count was determined by measuring turbidity according to the McFarland scale. For the experiments, the culture was brought to the necessary concentration. To confirm the amount of *P. aeruginosa* that was used in the experiment, aliquots of night culture (100 μL) were selected, a series of sequential dilutions was prepared, and 100 μL of diluted culture at each dilution was applied to a Petri dish with cetrimide agar. Petri dishes were cultivated at 37 °C. After 24 h of cultivation, the colonies were counted.

The bactericidal activity of the surfaces studied was analyzed using four protocols.

According to the first protocol, bare, superhydrophobic, and superhydrophilic copper plates were immersed in vials containing 15 mL of bacterial suspension with 10^7^ CFU/mL of *P. aeruginosa* PA103. Pure bacterial culture *P. aeruginosa* PA103 was used as a positive control. All samples were tested in triplicate. Vials were stored at room temperature or at 30 °C. In order to assess the antibacterial activity of bare, superhydrophobic, and superhydrophilic copper substrates submerged in bacterial suspension, after the prescribed time (1, 4, 24, 96, 120, 144, 168, 216 and 312 h), 0.1 mL of dispersion was taken from each vial and applied to a Petri dish containing sterile cetrimide agar. A Petri dish was placed in a thermostat for 24 h at 37 °C, then the colony-forming units (CFUs) were counted.

According to the second protocol, filter paper soaked in sterile saline was placed in Petri dishes to create wet conditions. The test plates were placed on the filter paper and 0.01 mL of bacterial suspension with 10^8^ CFU/mL of *P. aeruginosa* was applied on the test plate. The Petri dishes were covered with lids and left at room temperature. The bactericidal activity of bare, superhydrophilic, and superhydrophobic copper substrates in the second protocol was studied after 15 and 30 min and 1, 2, 3, 5, and 6 h. After the prescribed time, the samples of the coatings under investigation with the bacterial suspension were placed in vials containing 1 mL of sterile saline. Vials were shaken for 10 min at 250 rpm. A total of 100 μL of the dispersion was taken from the vials and applied to a Petri dish containing sterile cetrimide agar. To study the bacterial contamination of the plates, they were removed from the saline and transferred to 24-well tablets. In the 24-well tablets, 1 mL of sterile LB-broth was applied to the surface of the test samples and the tablets were incubated for 24 h at a temperature of 37 °C. After incubation, 100 μL quantities of broth from the wells with plates were placed in Petri dishes and distributed on the surface of the agar with the help of an L-shaped microbiological spatula. The dishes were placed in a thermostat for 24 h at 37 °C, and the number of CFU was counted after that.

The third protocol was used to determine the nature and duration of the antibacterial action of bare, superhydrophobic, and superhydrophilic copper substrates. According to it, a night culture of *P. aeruginosa* was brought to the desired titer in phosphate-buffered saline (PBS) and placed in quantities of 1 mL each on the studied plate samples in a 24-well tablet. The tablet was left for 24 h at room temperature. At the end of incubation, 100 μL of PBS from the wells was applied to Petri dishes containing a sterile nutrient medium (cetrimide agar). Petri dishes were placed in a thermostat for 24 h at 37 °C, and CFUs were then counted. In order to identify uncultivated forms of *P. aeruginosa,* after the incubation of the plates in PBS, 100 μL of the suspension from the wells was placed into 5 mL of sterile LB-broth and cultivated at 37 °C for 18 h under continuous shaking.

A fourth protocol was used to identify viable forms of *P. aeruginosa*. Accordingly, after incubating plates containing 0.01 mL of bacterial suspension for 3 h as described in the second protocol, the plates were placed in the well of a sterile 24-well tablet containing 1 mL of LB-broth and cultivated at 37 °C for 18 h. Bacterial cells were then sedimentated at 6000 rpm for 10 min, and the residue was used for further staining as described below.

### 3.3. Identification of Viable P. aeruginosa Forms

Viable forms of *P. aeruginosa* were identified by staining based on cell wall integrity and esterase activity using the Live/Dead Viability/Cytotoxicity Kit for bacterial cells (Invitrogen, Waltham, MA, USA).

## 4. Conclusions

It is clear today that long-term retention of hospital-acquired MDR pathogens on touch surfaces can be a source of cross-infection. Stainless steel and plastic are the most common materials for the manufacture of touch surfaces (door handles, switches, etc.), as they are corrosion resistant and are effectively treated with disinfectants. Recently, however, there has been a growing interest in the use of touch surfaces made of materials with antibacterial actions in hospitals, especially in ICUs and surgical departments. One of the best known such materials is copper, which, in low concentrations, can have an antibacterial effect [17]. In this study, we explored the antibacterial properties of superhydrophilic and superhydrophobic copper surfaces in relation to the reference strain and clinical isolates of *P. aeruginosa* with MDR.

We have shown that bare copper and superhydrophilic copper substrates have an antibacterial effect in 15 mL of *P. aeruginosa* PA103 bacterial suspension, but it is only manifested at room temperature. Experiments conducted at 30 °C reveal no antibacterial action for the studied surfaces. In general, the study of the antibacterial activity of bare, superhydrophobic, and superhydrophilic copper substrates placed in a liquid nutrient medium did not show reliable positive results, which may be due to a number of factors, including the large volume of the bacterial suspension, the small area of the studied coating, etc.

The optimal method for studying the antibacterial activity of the samples was to apply a bacterial culture in the amount of 0.01 mL directly to the studied surfaces. When this protocol was used, the antibacterial activity of bare copper and superhydrophilic copper surfaces was shown against both *P. aeruginosa* PA103 and *P. aeruginosa* clinical isolates with acquired MDR. The speed of the antibacterial response of the substrates varied with clinical isolate, and complete growth suppression occurred in 15 min to 2 h after the application of the bacterial culture.

However, it remained unclear whether the antibacterial action of the studied surfaces was temporarily suppressing the viability of *P. aeruginosa* or completely killing bacteria. Our experiments showed that when *P. aeruginosa* PA103 was cultivated in PBS with the addition of bare, superhydrophilic, and superhydrophobic copper substrates, growth was completely suppressed at room temperature for 24 h. However, when the PBS assay after contact with copper substrates was cultivated in a nutrient medium without metal for a further 24 h, two different scenarios were observed depending on the initial pseudomonas titer. It turned out that cell viability was completely suppressed at the initial culture titer of 10^5^ CFU/mL, whereas the growth of pseudomonas was restored in the assay with an initial titer of 10^6^ CFU/mL.

Staining with a specific dye was carried out to differentiate viable and non-viable cells. It has been shown that the studied surfaces have bactericidal rather than bacteriostatic effects, and, in previous experiments, bacterial populations recovered when they were exposed to favorable conditions due to the small number of surviving pseudomonas.

The obtained results allow the conclusion that upon contact with copper plates, pseudomonas dies, and uncultivated forms do not get formed with the possibility of their further propagation and transition to vegetative forms. A number of experiments have shown that bare copper and superhydrophilic copper substrates have effective antibacterial action against *P. aeruginosa*. Superhydrophobic copper substrates also showed antibacterial action; however, it was much less pronounced than for the superhydrophilic ones.

The observed effective suppression of *P. aeruginosa* isolates with MDR makes it possible to conclude that the use of superhydrophilic copper surfaces is promising for the manufacture of contact surfaces in healthcare units, where the risk of spread of and infection by hospital-acquired pathogens is extremely high.

## Figures and Tables

**Figure 1 ijms-25-00779-f001:**
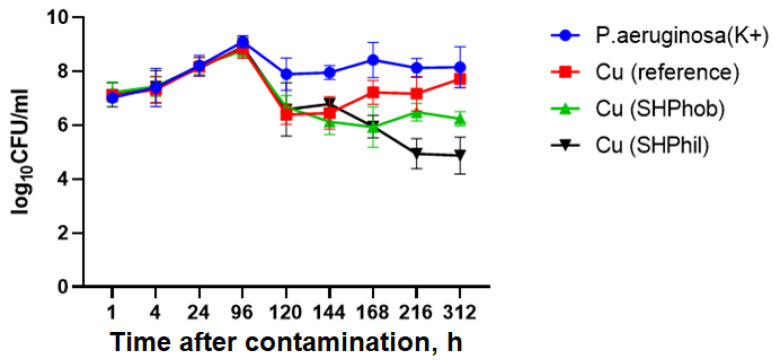
Evaluation of the antibacterial activity of superhydrophilic and superhydrophobic copper coatings against *P. aeruginosa* PA103 at room temperature after 1, 4, 24, 96, 120, 144, 168, 216, and 312 h. K+ denotes the positive control sample (vial with bacterial dispersion without any metal sample immersed).

**Figure 2 ijms-25-00779-f002:**
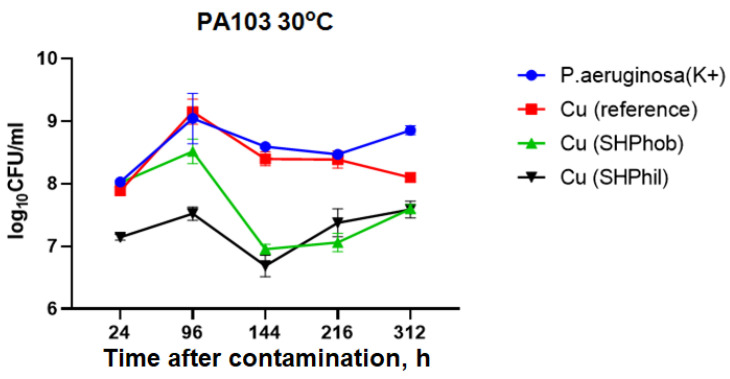
Estimation of the antibacterial activity of bare, superhydrophilic, and superhydrophobic copper coatings against *P. aeruginosa* PA103 at 30 °C after 24, 96, 120, 144, 168, 216, and 312 h.

**Figure 3 ijms-25-00779-f003:**
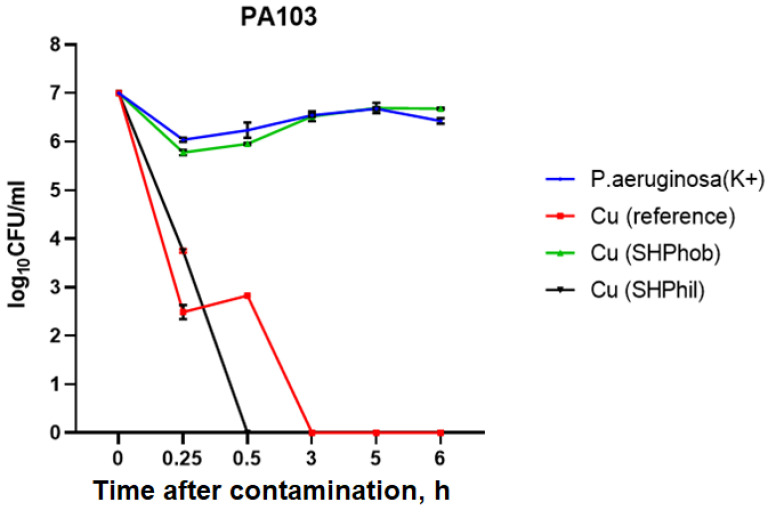
Evaluation of the antibacterial activity of bare, superhydrophilic, and superhydrophobic copper coatings against *P. aeruginosa* PA103 within 6 h of deposition of the bacterial culture to the plates.

**Figure 4 ijms-25-00779-f004:**
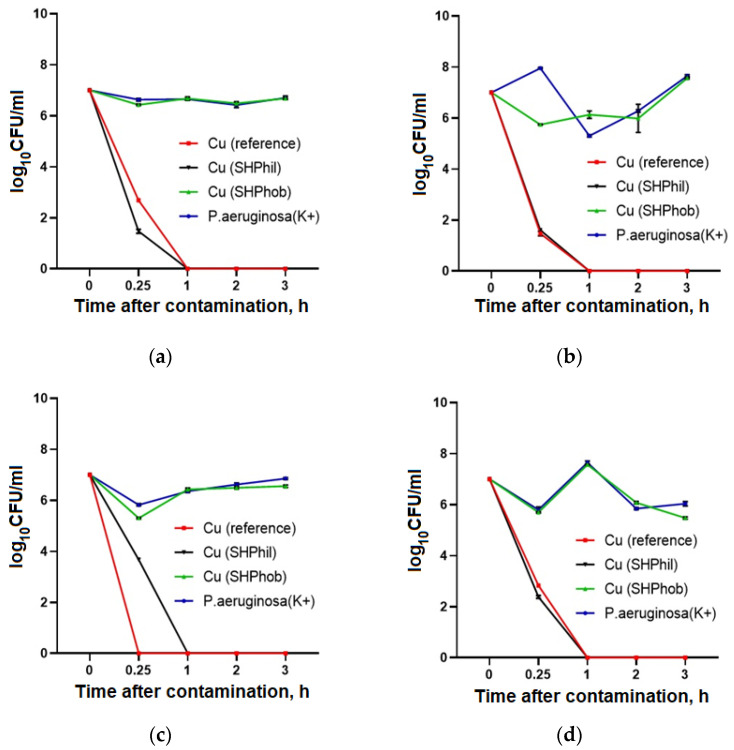
Evaluation of the antibacterial activity of bare, superhydrophilic, and superhydrophobic copper surfaces against *P. aeruginosa* isolates GIMC5019:PA52Ts1 (**a**), GIMC5016:PA1840/36/2015 (**b**), GIMC5021:PA52Ts17 (**c**), and GIMC5015:PAKB6/2014 (**d**), within 3 h after application of the bacterial culture to the plates.

**Figure 5 ijms-25-00779-f005:**
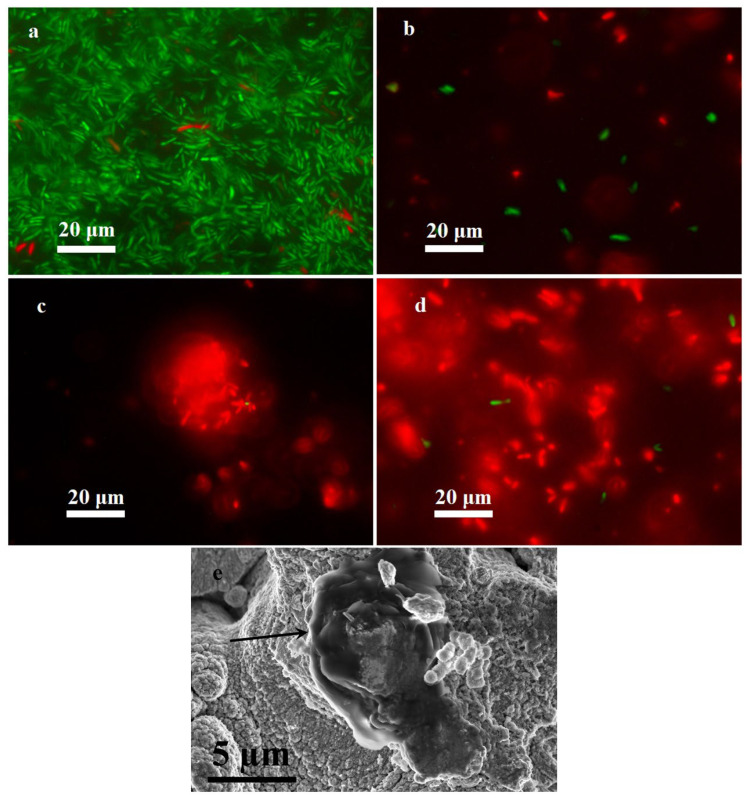
Identification of viable forms of *P. aeruginosa* PA103: (**a**) positive control sample, (**b**) bare copper substrate, (**c**) superhydrophilic copper substrate, (**d**) superhydrophobic copper substrate, and (**e**) scanning electron microscopy image of the cell with disrupted walls (indicated by an arrow) on a superhydrophilic surface. Viable forms of *P. aeruginosa* are colored green, and non-viable forms are colored red.

**Table 1 ijms-25-00779-t001:** Average values for cultured isolates after 24 h incubation with bare, superhydrophilic, and superhydrophobic copper for the *P. aeruginosa* PA103 strain.

Plate Samples	Initial Titer of Bacterial Culture (CFU/mL)	
	10^5^	10^6^
Cu (reference)	0	0
Cu (SHPhil)	0	0
Cu (SHPhob)	0	2.5 × 10^2^
Positive control	8 × 10^4^	6.7 × 10^6^

**Table 2 ijms-25-00779-t002:** Average values for cultured isolates from PBS after incubation with bare, superhydrophilic, and superhydrophobic copper substrates.

Plate Samples	Initial Titer of Bacterial Culture (CFU/mL)	
	10^5^	10^6^
Cu (reference)	0	3 × 10^8^
Cu (SHPhil)	0	3 × 10^8^
Cu (SHPhob)	2 × 10^8^	4 × 10^8^
Positive control	7 × 10^8^	1 × 10^9^

**Table 3 ijms-25-00779-t003:** Studied strains and isolates of *P. aeruginosa*.

Strain/Isolate Name	Source of Isolate	Susceptibility to Antibacterial Drugs	
		Sensitive	Resistant
*P. aeruginosa* PA103	sputum	Azlocillin Imipenem Tobramycin Gentamicin Levofloxacin	Ampicillin Chloramphenicol
*P. aeruginosa* GIMC5015:PAKB6/2014	bronchoalveolar lavage	Aztreonam Colistin Piperacillin/Tazobactam	Amikacin Amoxicillin/ Clavulanic acid Ampicillin Cefazolin Cefepime Cefoxitin Ceftazidime Ceftriaxone Ciprofloxacin Ertapenem Gentamicin Imipenem Meropenem Ticarcillin-Clavulanate Tigecycline TMP/SMX
*P. aeruginosa* GIMC5016:PA1840/36/2015	wound exudate	Colistin Piperacillin/Tazobactam	Amikacin Gentamicin Imipenem/Cilastatin Levofloxacin Meropenem Piperacillin Tobramycin Cefazolin Cefepime Cefoperazone/ sulbactam Cefotaxime Ceftazidime Ceftriaxone Ciprofloxacin
*P. aeruginosa* GIMC5019:PA52Ts1	tracheal aspirate	Colistin Amikacin Piperacillin/Tazobactam	Imipenem Meropenem Aztreonam Cefepime Ceftazidime Gentamicin Tobramycin Fosfomycin Ofloxacin Norfloxacin Levofloxacin Ciprofloxacin
*P. aeruginosa* GIMC5021:PA52Ts17	tracheal aspirate	Colistin Piperacillin/Tazobactam	Imipenem Meropenem Aztreonam Cefepime Ceftazidime Gentamicin Tobramycin Fosfomycin Ofloxacin Norfloxacin Levofloxacin Ciprofloxacin Amikacin

## Data Availability

All data are contained within the article.

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
