# Peer review of "Study of the Antibacterial Activity of Superhydrophilic and Superhydrophobic Copper Substrates against Multi-Drug-Resistant Hospital-Acquired Pseudomonas aeruginosa Isolates"

_ijms, 2024, doi:10.3390/ijms25020779_

Round 1

Reviewer 1 Report

Comments and Suggestions for Authors

The draft presents highly relevant research on the creation and practical application of superhydrophilic and superhydrophobic copper surfaces to combat dangerous hospital microorganisms. The experiment was structured logically; the authors stage-by-stage assessed the antimicrobial activity in comparison with control samples and untreated substrate. The results obtained are beyond doubt and are confirmed by the analysis of four research protocols and the identification of viable forms after cultivation. The article may be recommended for publication.

1) L59 “In order to prevent copper exposure to the body, it is possible to apply on copper the coatings with an extreme wettability.” How can creating extreme wetting coatings on copper substrates prevent copper exposure to the human body?

2) Why is there an antimicrobial effect when superhydrophobic substrates are completely immersed in a bacterial suspension, but when a drop is placed there is practically no effect?

3) From the results obtained it is clear that the original and superhydrophilic copper substrate exhibit the greatest antimicrobial activity. However, it is not entirely clear from the text which of these samples is most promising for further use?

4) Laser texturing creates a multimodal structure with a large surface area, does this affect antimicrobial activity?

Author Response

We thank the Reviewer for careful reading of our manuscript, positive evaluation and valuable comments and questions which certainly helped us to improve the manuscript.

In the text below, the Reviewer’s comments are given in blue font, our point-by-point replies in black, and changes made to the manuscript are highlighted with yellow color.

Reviewer 2 Report

Comments and Suggestions for Authors

The authors have demonstrated the antibacterial activity of superhydrophilic and superhydrophobic copper surfaces against the P. aeruginosa strain PA103 and its four different polyresistant clinical isolates with MDR. The authors have proven the promising potential of exploitation of superhydrophilic coatings in the manufacture of contact surfaces for healthcare units. Overall, this work can inspire more surface design ideas of copper-based materials for antibacterial application. Therefore, I would like to recommend this work to publish in International Journal of Molecular Sciences. Below are some comments for the authors.

1. In the introduction, more details of experimental designs of the study should be provided at the end of the introduction.

2. The unit of “ml” should be corrected to be “mL”.

3. To evaluate the properties of superhydrophilic and superhydrophobic copper surfaces, the data of contact angle should be provided.

4. As shown in Figure 5, to further demonstrate the disruption of the cell wall, the scanning electron microscopy (SEM) image can be used to check the disruption of the cell wall. This paper would be more impressive if the authors could provide SEM image with the disruption of the cell wall.

5. For the introduction “One way to combat the spread of nosocomial infections through prolonged exposure to contact surfaces may be to manufacture these surfaces from microbicidal materials”, more references could be cited to broaden the introduction.

https://doi.org/10.2147/IJN.S392081

Author Response

(The authors gave the same response as above.)
